# Genome Comparisons between *Botrytis fabae* and the Closely Related Gray Mold Fungus *Botrytis cinerea* Reveal Possible Explanations for Their Contrasting Host Ranges

**DOI:** 10.3390/jof10030216

**Published:** 2024-03-14

**Authors:** Klaus Klug, Pinkuan Zhu, Patrick Pattar, Tobias Mueller, Nassim Safari, Frederik Sommer, Claudio A. Valero-Jiménez, Jan A. L. van Kan, Bruno Huettel, Kurt Stueber, David Scheuring, Michael Schroda, Matthias Hahn

**Affiliations:** 1Department of Biology, University of Kaiserslautern-Landau, 67663 Kaiserslautern, Germany; klug@rhrk.uni-kl.de (K.K.); pkzhu@bio.ecnu.edu.cn (P.Z.); safari@rptu.de (N.S.); frsommer@bio.uni-kl.de (F.S.); davidscheuring@gmail.com (D.S.); m.schroda@rptu.de (M.S.); 2School of Life Sciences, East China Normal University, Shanghai 200241, China; 3Laboratory of Phytopathology, Wageningen University, 6708 PB Wageningen, The Netherlandsjan.vankan@wur.nl (J.A.L.v.K.); 4Max Planck Genome Centre Cologne, Max Planck Institute for Plant Breeding Research, 50829 Cologne, Germany

**Keywords:** *Botrytis*, host range, host specificity, transposable elements, gene degeneration, pathogen evolution, secondary metabolite gene cluster

## Abstract

While *Botrytis cinerea* causes gray mold on many plants, its close relative, *Botrytis fabae*, is host-specifically infecting predominantly faba bean plants. To explore the basis for its narrow host range, a gapless genome sequence of *B. fabae* strain G12 (BfabG12) was generated. The BfabG12 genome encompasses 45.0 Mb, with 16 chromosomal telomere-to-telomere contigs that show high synteny and sequence similarity to the corresponding *B. cinerea* B05.10 (BcB0510) chromosomes. Compared to BcB0510, it is 6% larger, due to many AT-rich regions containing remnants of transposable elements, but encodes fewer genes (11,420 vs. 11,707), due to losses of chromosomal segments with up to 20 genes. The coding capacity of BfabG12 is further reduced by nearly 400 genes that had been inactivated by mutations leading to truncations compared to their BcB0510 orthologues. Several species-specific gene clusters for secondary metabolite biosynthesis with stage-specific expression were identified. Comparison of the proteins secreted during infection revealed high similarities, including 17 phytotoxic proteins that were detected in both species. Our data indicate that evolution of the host-specific *B. fabae* occurred from an ancestral pathogen with wide host range similar to *B. cinerea* and was accompanied by losses and degeneration of genes, thereby reducing its pathogenic flexibility.

## 1. Introduction

Plant diseases caused by fungi are among the greatest threats to global crop production and human food supply. Plant pathogenic fungi are generally highly adapted organisms and often show high degrees of host specificity. For example, rust and powdery mildew fungi, which belong to the economically most important plant pathogens, are completely dependent on their host for growth and reproduction, and often infect only one of few plant species. The molecular basis of host specificity is not yet well understood. An important role is played by secreted fungal effector proteins that promote infection by the pathogen. On the other hand, plants containing resistance proteins that recognize specific effectors can build up a strong immune response and prevent infection. In addition, the ability of some fungi to release phytotoxic metabolites and to overcome the chemical defense of their host plant are further determinants of infection, which can also affect host specificity [1]. Understanding the determinants of host specificity is of great agronomic importance, because evolutionary pressure can lead to sudden host jumps, as exemplified by the appearance in the 1980s in Brazil of a wheat pathotype of the rice blast fungus *Magnaporthe oryzae*, which is causing increasing damage in many wheat-producing countries [2].

The genus *Botrytis*, established in 1729, is among the first described genera of the fungi. Within the Ascomycota phylum, it belongs to the class Leotiomycetes, family Sclerotiniaceae. Botrytis species can be grouped into two phylogenetic clades: clade 1, including *B. cinerea* and several other species infecting mainly dicot host plants, and clade 2, comprising more than 30 species infecting monocots and dicots [3,4,5]. Most *Botrytis* species are necrotrophic pathogens that rapidly kill the host tissue [3]. The best-studied *Botrytis* species is *B. cinerea*, which has been reported to attack more than 1000 plant species [6]. It infects a large variety of important fruit, vegetable and flower crops worldwide, and is considered one of the economically most important fungal plant pathogens [7]. Another species with a wide host range, *B. pseudocinerea*, is phenotypically indistinguishable from *B. cinerea* and often found on the same host plants [8,9]. In contrast, most of the other *Botrytis* spp. are host-specific and infect few, often related, plant species. The molecular basis for the extraordinarily wide host range of *B. cinerea*, and for the different host ranges of *Botrytis* species in general, is poorly understood.

*Botrytis fabae* Sard. is among the closest relatives of *B. cinerea*, but infects few legume genera, including *Vicia*, *Lens* and *Trifolium*. It has been described as the major pathogen of broad bean (*Vicia faba*) and induces typical chocolate spot lesions [10]. *B. fabae* broad beans are also infected by other *Botrytis* species, such as *B. cinerea*, *B. pseudocinerea* and *B. fabiopsis* [9,10,11,12,13]. Phenotypically, *B. fabae* is clearly distinguishable from *B. cinerea* based on reduced sporulation on artificial media, larger conidia, and smaller sclerotia [10,14]. In response to infection, *Vicia faba* produces the antifungal furanoacetylene phytoalexins wyerone and wyerone acid [15]. Compared to *B. cinerea*, *B. fabae* shows a higher tolerance to these phytoalexins [16], possibly as a result their conversion to the less toxic wyerone derivate [17].

Infection of *B. cinerea* is associated with the release of lytic enzymes that breach and degrade the walls and other components of the host cells [18,19,20,21]. Host killing by necrotrophic fungi has been shown to occur by induction of the hypersensitive response, a defense-related form of programmed cell death (PCD) effective against many pathogens [22]. *B. cinerea* secretes a large number of cell death-inducing proteins (CDIPs) and phytotoxic metabolites, several of which have been described as virulence factors [23,24]. Sequential deletion of CDIPs and phytotoxic metabolites resulted in markedly reduced virulence on different host tissues, but none of the individual CDIPs appeared to play a major role [25]. Thus, *Botrytis*-induced necrosis seems to be the combined result of PCD induction and cell wall degradation [22,25,26]. Tissue acidification and the secretion of organic acids have been shown to accelerate lesion formation [27]. In contrast to most plant pathogenic fungi, little is known about effector proteins that suppress the host defense. *B. cinerea* releases small RNAs that suppress host defense gene expression in Arabidopsis and tomato [28], but their biological role has recently been questioned [29].

The first genome sequence of *B. cinerea* was published in 2011 [30]. By using a combination of Illumina and PacBio sequencing technologies and optical mapping, a finished genome sequence was obtained for *B. cinerea* strain B05.10 [31]. In the meantime, genomes of another 15 *Botrytis* species have become available. They revealed a heterogeneous pattern of secondary metabolite key enzymes, and a conserved genome structure with 16 core chromosomes [32,33]. The contrasting host ranges of *B. fabae* and *B. cinerea*, despite their close phylogenetic relationship, make these species particularly attractive for genome comparisons and facilitate the discovery of evolutionary mechanisms underlying their physiological diversification. We therefore generated a gapless, finished genome sequence of *B. fabae* strain G12, with manually edited gene models supported by RNA sequencing data. Comparison with *B. cinerea* B05.10 revealed very high similarity of both genomes and their encoded genes, but also distinct differences regarding their coding capacities and the number of species-specific genes.

## 2. Materials and Methods

### 2.1. Fungal Cultivation, Infection Assay and Transformation

*Botrytis fabae* strain G12 was isolated in Greece from an infected *Vicia faba* leaf. *B. cinerea* B05.10 is a commonly used wild-type strain. *B. cinerea* was cultivated on malt extract agar [25] and *B. fabae* on medium X agar (10% sucrose, 3% NaNO_3_, 1.8% agar, 20% homogenized Vicia fabae leaves; pH 5.5) [8] to improve sporulation. Agar plates were incubated at 20–22 °C under fluorescent light. Transformation of *B. fabae* was performed using a published protocol [25], with a fragment amplified from plasmid pNAH-OGG containing expression cassettes for GFP and hygromycin resistance [32].

### 2.2. DNA Sequencing, Genome Assembly and Annotation

DNA for genome sequencing of *B. fabae* was isolated by grinding 8 × 107 conidia in liquid nitrogen using a mortar, followed by the yeast protocol of a blood and cell culture DNA kit mini [Qiagen, Hilden, Germany]. The DNA was fractionated using BluePippin [Sage Science, Inc., Beverly, MA, USA] to yield fragments of >7 kb size. Sequencing was performed with PacBio RS II in six SMRT cells, resulting in reads of up to 60 kb and 90-fold genome coverage. Assembly of the reads was performed with de novo assemblers HGAP2 and HGAP4 [Pacific Biosciences, Menlo Park, CA, USA]. The closing of two remaining gaps in the assemblies was achieved by PCR and Sanger sequencing using primers in the flanking regions of the gaps and “within-gap” primers derived from *B. cinerea* genome sequences that corresponded to the missing sequences in the *B. fabae* genome. Low-coverage (ca. 30×) sequencing of *Botrytis fabae* 11002 and *B. pseudocinerea* VD110 genomes was achieved using Illumina HiSeq2000. Repeat-induced point mutations (RIP) were analyzed using RIPCAL 2.0 [33]. For annotation of the *B. fabae* genome, the gene prediction programs Augustus [34], GlimmerHMM [35] and GeneMark-ES [36] were run and later integrated by Evidence Modeler EVM [37]. A training set based on about a thousand manually curated *B. cinerea* gene models [38] was used to train Augustus. Extensive manual curation of the gene models was performed after loading the gff3 file into Web Apollo [39] together with RNA sequencing data to verify the gene models and to identify the correct exon–intron junctions. Because of the high similarity (>97%) of *B. fabae* and *B. cinerea* genes, gene models poorly supported by RNAseq were taken from the *B. cinerea* annotation if possible. Identification of TE sequences was conducted with RepeatMasker. Remnants of TE coding sequences in AT-rich regions were identified additionally by similarity to sequences in Genbank/NCBI.

### 2.3. RNA Isolation and Sequencing

*B. fabae* strain G12 RNA for sequencing was isolated from: (1) conidia harvested from 12 d cultures on medium X agar; (2) non-sporulating aerial mycelium grown for 4 days under ambient light, scraped from plates; (3) 12 h-old germlings grown on petri dishes coated with 10 µg/mL apple wax in GB5 medium with 10 mM glucose; (4) immature sclerotia formed on malt extract agar after 14 days’ incubation in the dark; and (5) infected *Vicia faba* leaves 48 h after inoculation. Total RNA was isolated using the NucleoSpin RNA Plant Mini Kit [Macherey-Nagel, Düren, Germany]. After enrichment of poly(A)-RNA and reverse transcription, Illumina sequencing was performed using HiSeq3000 for 150 bp unpaired reads. RNAseq data were evaluated using the CLC genomics workbench (Qiagen), version 6.9, after reference alignment with the *B. fabae* G12 genome sequence. Expression values of mapped reads were normalized using Baggerly’s test. Genes with RPKM values >2 were considered significantly expressed.

### 2.4. Confocal Microscopy

For image acquisition, a Zeiss LSM880 AxioObserver confocal laser scanning microscope equipped with a Zeiss C-Apochromat 40×/1.2 W AutoCorr M27 water-immersion objective was used (INST 248/254-1). Fluorescent GFP signals (excitation/emission 488 nm/500–553 nm) and RFP signals (excitation/emission 543 nm/580–718 nm) were processed using Zeiss software ZEN 2.3 or ImageJ 1.54.

### 2.5. Secretome Analysis

On planta, secretomes of *B. fabae* and *B. cinerea* were obtained from *Vicia faba* leaves as described previously [25,32]. Shortly after, inoculation droplets from infected leaves (48 h.p.i.) were collected and the proteins purified and analyzed by MS/MS using an ABSciex Triple-TOF 6600 mass spectrometer. Data evaluation was performed with four biological replicate samples for both species on the Perseus computational platform using a *t*-test, with an FDR of 0.05 and S0 value of 0.1.

## 3. Results

### 3.1. Botrytis fabae and B. cinerea Show Different Modes of Invasion into Faba Bean Leaf Tissue

Despite their close evolutionary relationship, *Botrytis fabae* and *B. cinerea* differ considerably in their vegetative growth, sporulation efficiency and sclerotia formation (Figure 1A). On leaves of faba bean (*Vicia faba*), *B. fabae* isolates are usually more virulent than *B. cinerea* isolates [9]. After inoculation of faba bean leaves with conidial suspensions, *B. fabae* strain G12 formed larger lesions than the *B. cinerea* strain B05.10 (Figure 1B). To compare their infection process microscopically, GFP-labeled strains of *B. cinerea* and *B. fabae* were used. *B. cinerea* formed expanding lesions with hyphae that were growing outward mostly radially (Figure 1C). As observed by the loss of red chlorophyll autofluorescence, a front of dead mesophyll cells ahead of these hyphae was visible, indicating the release of phytotoxic compounds from the expanding mycelium. In contrast, hyphae of lesions caused by *B. fabae* appeared to be less ordered in their growth direction and induced less extensive plant cell death ahead of the hyphal tips (Figure 1C,D). These observations indicate significant differences in the mode of proliferation of necrotrophic hyphae between *B. fabae* and *B. cinerea*.

### 3.2. A Gapless Genome Assembly of B. fabae Reveals High Synteny with B. cinerea

High-molecular-weight DNA of *B. fabae* strain G12 (BfabG12) was isolated from conidia and subjected to genome sequencing using PacBio single-molecule sequencing technology, with a sequencing depth of 90-fold the genome size. Alignment of the assemblies resulted in 18 contigs, which could be aligned to the 16 large chromosomes present in the *B. cinerea* strain B05.10 (BcB0510). Numbering of the *B. fabae* chromosomes was carried out according to those of BcB0510. Two remaining gaps, 533 bp (chrom. 7: 2,434,373–2,434,906) and 12 kb (chrom. 12: 1,539,166–1,551,218) in size, were closed by a single PCR for Chr7 and by primer walking and sequencing for Chr12, using the highly homologous BcB0510 sequence as template (Appendix A). Complete sequence coverage of all BfabG12 chromosomes was confirmed by the detection of telomeric repeats [(TTAGGG)n] on both ends of each of the 16 chromosomes. A comparative map of chromosomes from BcB0510 and BfabG12 is shown in Figure 2. The rDNA region located close to a telomere in BfChr4 was found to contain at least five rDNA repeats, each 9250 bp in length, but could not be resolved further by the sequencing reads. In BcB0510, the rDNA region was determined by optical mapping to 580 kb, corresponding to >60 rDNA repeats [29], and a similar number was estimated for BfabG12. The genomes of the two species revealed a high degree of synteny. Overall, 12 inversions between 10 and 400 kb, 18 small inversions (<10 kb), and two translocations were detected between the two genomes (Figure 2; Appendix A). The larger translocation (125 kb; 56 kb in BcB0510), including a cluster of 13 genes for the biosynthesis of the phytotoxic polyketide botcinic acid, which is located at the left end of Chr1 in *B. cinerea* [29,40], was found at the right end of BfChr10. Two small chromosomes of BcB0510, BcChr17 (247 kb) and BcChr18 (209 kb) are missing in BfabG12, but a region corresponding to 15 kb of BcChr17 (29 kb in BfabG12) containing five genes is located at the right end of BfabChr4. In BcB0510 chromosomes, regions have been predicted to represent centromeres, based on their low GC content, their size and the lack of annotated genes [29]. Similar regions at corresponding locations were found in each of the BfabG12 chromosomes (Appendix A).

### 3.3. The B. fabae G12 Genome Is Expanded Compared to B. cinerea B05.10 Due to Enriched AT-Rich Regions Containing Remnants of Transposable Elements

A significant expansion of the BfabG12 genome relative to BcB0510 was observed. Assuming a similar size of the rDNA repeats in *B. fabae* as in *B. cinerea*, the size of the BfabG12 genome is about 45.50 Mb, compared to 43.23 Mb of BcB0510. This expansion was observed for each of the 16 chromosomes and accompanied by a consistently lower GC content within BfabG12 chromosomes (Table 1). Scanning of the GC content along the chromosomal DNA revealed that in addition to the putative centromeres shared with BcB0510, all BfabG12 chromosomes are enriched in AT-rich regions, which are up to 95 kb in size (Table 1, Appendix A). The majority of these regions were specific for BfabG12, whereas fewer and shorter AT-rich regions were found in BcB0510 (Appendix A). AT-rich regions in fungal genomes have been shown to originate from transpositions of mobile elements (TE), followed by repeat-induced point mutations (RIP), a premeiotic defense system against repeated DNA that results in C-to-T mutations in duplicated sequences [33]. This mechanism was confirmed for most of the AT-rich regions of BfabG12 by two lines of evidence. First, remnants of coding sequences with TE origin were detected in the majority of the larger AT-rich regions (see below). Secondly, RIP-specific indices in all AT-rich regions revealed an increased frequency of C-to-T and C-to-A mutations (Figure 3). In *Leptosphaeria maculans*, AT-rich regions have been found to contain newly evolved genes encoding secreted effector proteins [41]. Inspection of the AT-rich regions in the *B. fabae* and *B. cinerea* genomes, however, revealed no evidence of the presence of novel genes. In contrast, several AT-rich regions were found next to deletions of *B. fabae* genes (e.g., homologues of Bcin06g01370-01390) or next to genes that are absent in *B. cinerea* (e.g., Bfab09g05695). Furthermore, some AT-rich regions were parts of inversions between the two genomes (Appendix A).

Transposable elements (TEs) found in *B. cinerea* belong to long terminal repeat retroelements (LTR-RE) of the Gypsy and Copia superfamilies, long interspersed nuclear (retro)elements (LINEs), and terminal inverted repeat (TIR) DNA transposons [42,43]. The Gypsy retroelement Boty [44] and the TIR transposon Flipper [45] have often been used for population studies, and their existence in *B. cinerea* field strains was reported to be correlated with differences in their pathogenicity and seasonal changes in their abundance [46,47,48]. When AT-rich regions were screened for the presence of TE gene sequences, no intact genes were found, as expected, but in 358 of 483 AT-rich regions in BfabG12, genes belonging to different classes of TE were detected, with clear evidence for RIPing and multiple premature stop codons (Appendix A). To identify active transposons in BfabG12, we searched for genes encoding intact Gag-Pol polyproteins (of LTR-RE) and transposases (of DNA transposons). BfabG12 contains five such Gag-Pol gene copies of the Gypsy element Boty (BcB0510: 25), one copy belonging to the Ty1-Copia element (BcB0510: five), and five copies of genes for transposases belonging to two Fot1-like elements related to Flipper (BcB0510: 15) (Appendix A). For all intact TE copies, the flanking LTR sequences were also detected.

### 3.4. Lower Gene Content of B. fabae Genome by Loss of Segments Carrying Several Genes

Annotation of the BfabG12 genes was started with ab initio predictions using Augustus trained with a set of confirmed *B. cinerea* gene models [38]. Transcript-based support for the gene models was obtained by RNA sequencing from different developmental stages of BfabG12 (see below). The BfabG12 genome sequence and preliminary gene models were loaded into the Web Apollo browser together with the RNA reads. BcB0510 gene models were also loaded, whichd allowed comparison of the corresponding gene models. Manual curation was performed for every BfabG12 gene model, if necessary. Because of the high degree of synteny between both genomes, all *B. fabae* genes were named according to their *B. cinerea* orthologues, e.g., Bfab09g04540 vs. Bcin09g04540. BfabG12 and genes that are missing in BcB0510 were assigned numbers (e.g., Bfab09g04545 and Bfab09g04546) analogous to their position (e.g., Bfab09g04540, Bfab09g04545, Bfab09g04546, Bfab09g04550). BfabG12 genes with BcB0510 orthologues at a different position were named according to their BcB0510 orthologue, with an extension indicating their chromosomal location, e.g., Bfab10g0100060, an orthologue of Bcin01g00060, which is located on chromosome 10.

Annotation of the BfabG12 genome revealed 11,420 predicted genes, compared to 11,707 annotated genes in BcB0510 [29] (http://fungi.ensembl.org/Botrytis_cinerea, accessed on 1 March 2024; Table 2). Comparison with BcB0510 revealed 97.6% nucleotide identity for 11,082 BfabG12 genes that have orthologues in BcB0510. This high value confirms the close evolutionary relationship between both species. The different gene counts result mainly from 471 (426 intact) BcB0510 genes that are missing in BfabG12 (Appendix A), compared to only 148 (129 intact) BfabG12 genes absent in BcB0510 (Appendix A). Closer inspection of the strain-specific genes revealed that 219 (51.4%) of the BcB0510-specific genes but only 29 (22.5%) of the BfabG12-specific genes encode proteins with conserved domains or predicted functions. Furthermore, in contrast to 379 (80.5%) of the BcB0510-specific genes that are absent on the DNA level in BfabG12 (Appendix A), only 31 (21.4%) of the BfabG12-specific genes are absent in BcB0510 (Appendix A). Of the BcB0510 genes that are missing in BfabG12, 308 are clusters of two to twenty genes (Figure 4), and 136 are located at chromosomal ends or in small chromosomes. In contrast, only 28 of the 129 BfabG12 genes lost in BcB0510 are parts of clusters of fewer than five genes (Figure 4), and none of them is located at chromosomal borders.

To test to what extent the results of the genome comparisons can be generalized to the species level, we included draft genome sequence data of *B. fabae* strains 11002 (this work), Bf611 and Bf612 [12], the two *B. cinerea* strains T4 [38] and DW1 [49], *B. pseudocinerea* strain VD110, *B. calthae* strain MUCL2830 (both this work) (Appendix A), and eight other *Botrytis* species [30]. Using BLAST homology searches, the genomes of these species were screened for the presence or absence of genes that were specific for either BfabG12 or BcB0510. Of the 145 BfabG12-specific genes, 128 were also found in the low-coverage sequenced genomes of three other *B. fabae* strains, and 71 of them were absent in at least one of the three *B. cinerea* strains. Conversely, of the 476 BcB0510-specific genes, 185 were present in the three *B. cinerea* strains and absent in all *B. fabae* strains sequenced (Appendix A), thus confirming the different coding capacity of the two species. Remarkably, 150 of the 185 *B. cinerea*-specific genes that are missing in *B. fabae* were present in the broad host range species *B. pseudocinerea* strain VD110, compared to only 69 genes in the host-specific *B. calthae* strain MUCL2830 (Appendix A).

Genes missing in either BfabG12 or BcB0510, but present as orthologues in most other *Botrytis* strains or species, have likely been lost by phylogenetically recent events, and were classified as losses. In contrast, genes found only in one strain or species but not in any related species are referred to as “unique” genes. Irrespective of their origin (e.g., by acquisition via horizontal gene transfer, gene duplication followed by diversification, or losses in most related species), unique genes are of particular interest because they might confer specific functions and explain host specificity. While the majority of strain-specific genes resulted from losses in the other strain, we identified 14 unique BfabG12 genes and 93 unique BcB0510-specific genes that are absent in most or all other *Botrytis* species (Appendix A).

### 3.5. The Coding Capacity of B. fabae Is Further Reduced by the Degeneration of Genes

In addition to strain-specific genes, a similar number of genes were found to be truncated and probably nonfunctional in either BfabG12 or BcB0510. For identification of truncated genes, only gene sequences were considered that had reliable base counts and were verified by RNAseq data. In BfabG12, the predicted proteins from 401 genes were at least 10% (set as threshold) smaller than their BcB0510 orthologues (Appendix A: Bfab trunc vs. Bcin), whereas proteins from only 72 genes were ≥10% smaller in BcB0510 than their BfabG12 orthologues (Appendix A: Bcin trunc vs. Bfab). The size reductions were interpreted as signs of degeneration and concomitant loss of function of the affected genes. These truncated genes further reduce the functional coding capacity of BfabG12 relative to BcB0510. To analyze the mutations leading to gene degeneration, homologous coding sequences of the two strains were compared. In 170 out of 401 BcB0510 genes (truncated in BfabG12; Appendix A) and in 29 out of 72 BfabG12 genes (truncated in BcB0510; Appendix A), individual mutations leading to protein truncations could be identified. These mutations comprised 67 point mutations, 63 single-nucleotide frameshift mutations and 40 deletions. Several genes with truncated homologues in the other strain also encoded truncated proteins based on comparison of their sizes with the best homologues in other fungi (Appendix A). The total number of intact, strain-specific genes was thus 786 for BcB0510 and 191 for BfabG12. After subtraction of those genes that were not significantly expressed (RPKM < 2) in any of the five developmental stages analyzed, the total numbers of intact and significantly expressed strain-specific genes were 569 (BcB0510) and 126 (BfabG12) (Table 2).

### 3.6. Transcriptome Analysis Reveals Low Expression of Many Strain-Specific Genes

To obtain evidence for the expression of strain- and species-specific genes, RNAseq data were generated from the following developmental stages of BfabG12: resting spores, germlings, vegetative mycelium, sclerotia and infected bean leaves. Compared to the average of all genes, gene expression levels of strain-specific genes were significantly lower. Furthermore, the expression of truncated genes was lower than their non-truncated homologues in the other strain. Similarly, the fraction of expressed genes among the strain-specific and truncated genes was lower compared to all genes (Appendix A).

### 3.7. Several B. fabae Genes Contain Introns with Non-Canonical Splice Junctions

Introns are non-coding sequences of eukaryotic genes that are excised after transcription from the pre-mRNA by the spliceosome. The border regions of introns are conserved in all eukaryotes, including fungi [50]. While most introns contain canonical 5′-GT and 3′-AG borders (GT-AG introns), a few genes contain introns without canonical borders. For human genes, non-canonical splicing has been suggested to be a source of new transcripts during evolution [51]. Among 22,877 introns identified in the 11,420 BfabG12 genes, the consensus sequence of most introns is GTaaGT for the 5′-border and CAG or TAG for the 3′-border. In 1.12% of introns, a minor consensus 5′-border was GCaaGT (Appendix A). These data are similar to other eukaryotes [50]. Sixteen BfabG12 genes were detected that contain RNAseq-supported introns with one or two non-canonical border sequences (Appendix A). Altogether, 9 5′-borders and 14 3′-borders with non-canonical sequences were identified. In BcB0510, four genes had different splice junctions, and one gene with non-canonical 3′-border in BfabG12 (Bfab03g06466) was missing in all other *Botrytis* species except *B. aclada*. The non-canonical border sequences were variable and could not be unified with a new consensus, except for six 5′-borders that shared the consensus “ATANGT.” Inspection of the 16 BfabG12 genes containing non-canonically spliced introns did not provide evidence for any special roles. Some of these genes encode conserved proteins (e.g., CTP synthase, vacuolar ATP synthase subunit E), whereas others encode hypothetical proteins (Appendix A).

### 3.8. B. fabae Has Reduced Capacity for Secondary Metabolite (SM) Synthesis, but Contains Two Stage-Specifically Regulated SM Clusters That Are Truncated in B. cinerea

Compared to BcB0510, BfabG12 lacks six SM-biosynthesis key enzymes and contains six enzymes as truncated copies. Conversely, BbabG12 contains three key enzymes that are missing or truncated in BcB0510 (Table 3). Of the approximately 44 SM potentially synthesized by BcB0510, 11 cannot be produced by BfabG12 because of lack or truncation of SM key enzymes, and another three because three to four genes of the associated clusters are missing (Table 3 and Appendix A). As noted above, the botrydial cluster, containing seven genes, is missing in *B. fabae* except for one remaining gene [52]. From a *B. cinerea* gene cluster involved in synthesis of the plant hormone abscisic acid [53], three genes are missing in *B. fabae*. A *B. fabae*-specific putative cluster for polyketide biosynthesis consists of three weakly expressed genes (Bfab09g04545/50/55) encoding an oxidoreductase, a PKS, and a transcription factor, respectively. Three unique *B. fabae* genes (Bfab03g06465/66/67) and a gene encoding a PKS (Bfab03g06470) belong to five clustered genes with sclerotium-specific expression (Figure 5A). Truncated versions of the PKS are encoded in *B. cinerea* strains T4 and DW1, but not in any other *Botrytis* species. Therefore, the cluster could be involved in the biosynthesis of a *B. fabae*-specific polyketide during sclerotia formation. Another cluster of six genes in *B. fabae*, of which two are missing and one is truncated in *B. cinerea*, was strongly upregulated in planta (Figure 5B). One of the six genes encodes a fatty acid desaturase, indicating that the cluster could be involved in the synthesis of a lipid-containing metabolite.

### 3.9. Functional Classification of Strain-Specific Genes

Strain-specific genes encoding proteins with predicted domains or functions, including 54 BfabG12-specific genes and 406 BcB0510-specific genes, were grouped into functional categories (Table 4). Cytochrome P450 monooxygenases (CytP450) and drug efflux transporters can contribute to synthesis and export of endogenous SMs or in detoxification of plant defense compounds. Compared to seven CytP450 and four transporters specific for BfabG12, 22 CytP450 and 34 efflux transporters are encoded only in BcB0510. BcB0510 contains 36 transcription factors that are missing in BfabG12, but only 3 BfabG12-specific transcription factors, indicating a higher regulatory capacity in *B. cinerea*. Similarly, BcB0510 encodes 27 carbohydrate-active enzymes and 31 secreted proteins which are missing in BfabG12, in contrast to only 6 BfabG12-specific genes in each of these categories (Appendix A).

### 3.10. Analysis of Gene Clusters Specific for B. cinerea

Among the 426 strain-specific BcB0510 genes, 308 (72.3%) are part of gene clusters that are missing in *B. fabae* (Appendix A). In sum, 181 of these genes are located at the ends of Bcin chromosomes, indicating that their loss in *B. fabae* might have occurred during chromosomal replication. Nine clusters of *B. cinerea*-specific genes showed stage-specific expression of at least three genes (Appendix A). Five of these clusters showed sclerotia-specific expression, three clusters in planta-specific expression, and the cluster for botrydial synthesis (Bcin12g06380-06430) mycelium-specific expression. A dominance of sclerotia-specifically expressed genes was observed for all *B. cinerea*-specific genes, compared to 48 genes upregulated in planta and 23 genes or less in other developmental stages (Appendix A).

### 3.11. Comparison of the on Planta Secretomes of B. fabae and B. cinerea

BfabG12 and BcB0510 encode similar numbers of predicted secreted proteins. There are only 5 BfabG12-specific and 23 BcB0510-specific genes for potentially secreted proteins with significant expression (Appendix A). Nine of the BcB0510 genes are upregulated in planta. Three of them are highly expressed in infected tissue (RPKM > 100), and two adjacent genes (Bcin01g03900/10) encode small, cysteine-rich proteins of 87 aa and 66 aa, which are both predicted to be effector proteins by EffectorP [54] (Appendix A). Together with a third gene (Bcin01g03890) encoding a hypothetical protein, they are located in a ~40 kb AT-rich region. Homologous genes are found only in *B. tulipae* and *Stagonospora* sp., but the clusters are different (Appendix A).

Proteins that are secreted by BfabG12 and BcB0510 during necrotrophic infection were compared by MS/MS analysis. For isolation of the on planta secretomes, inoculation droplets (cf. Figure 1B) were collected 48 h.p.i. from inoculated *V. faba* leaves, and the proteins released into the droplets were analyzed by LC-MS/MS [25]. The secretomes of BfabG12 and BcB0510 showed equal protein concentrations (8.44 ± 1.97 µg/mL and 8.38 ± 1.94 µg/mL, respectively) and revealed a similar overall composition (Figure 6; Appendix A): they shared 359 homologous proteins, 43 and 62 proteins were found exclusively or enriched in the BcB0510 secretome, respectively, and 30 (26) proteins were found exclusively or enriched in the BfabG12 secretome. These data show a slightly higher number of secreted proteins in the *B. cinerea* secretome. *B. cinerea* has been shown to secrete many phytotoxic proteins (CDIPs), which were shown to contribute in a quantitative manner to necrotrophic infection [23]. Homologues of 17 CDIPs that were detected in the secretome of *B. cinerea* were also found in the *B. fabae* secretome (Appendix A). Five CDIPs (Xyn11A, XYG1, CDI2, PG2, and IEB1) were found to be significantly more abundant in the *B. cinerea* secretome (Figure 6).

## 4. Discussion

In this study, we present a gapless, fully annotated genome sequence of *B. fabae* strain G12, which may be considered finished. This was achieved by using PacBio sequencing technology only. This approach was successful with the available, highly similar BcB0510 genome sequence and because of the low number of repeated sequences. High-resolution fungal genome sequencing has been made possible by long-read sequencing, often combined with high-coverage Illumina sequencing and optical mapping, as demonstrated for *Fusarium graminearum* [55], *Verticillium dahliae* [56], *B. cinerea* [29], *Colletotrichum higginsianum* [57], and *Zymoseptoria tritici* [58]. The benefits of complete chromosomal assemblies of fungal genomes have been demonstrated, which allow much deeper insights into their structure, evolution and function [59]. In our study, this was exploited for in-depth comparative analyses of AT-rich regions, rearrangements, and the dynamics of potential gains, losses and degeneration of genes and gene clusters between the two species. Chromosomal synteny between BfabG12 and BcB05.10 is very high, interrupted by only 12 inversions >10 kb, two translocations, and several deletions, most of them in *B. fabae*. The small chromosomes BcChr17 and BcChr18 are missing in *B. fabae*, but homologues for five BcChr17 genes were found at the end of BfabChr4. *B. cinerea* strains contain variable numbers of small chromosomes, as shown by karyotype analysis and genome sequencing [29,60]. A remarkable conservation of 16 core chromosomes has been reported in all *Botrytis* species and the related white mold pathogen *Sclerotinia sclerotiorum* [31,61], unlike the genomes of other fungi, such as *Fusarium* spp., which have variable numbers of core chromosomes, besides the occurrence of small non-core chromosomes [62].

The BfabG12 genome is expanded by ~5% relative to BcB0510, resulting from more extended AT-rich regions, which make up ~14% of the total genome size, compared to 4.9% in BcB0510 (Table 1). Variable amounts of AT-rich DNA were also found in *B. cinerea* strains B05.10 and T4 [28] and other *Botrytis* spp., ranging in size from 6.5% in *B. galanthina* to 27% of the whole genome in *B. narcissicola* isolates [30]. AT-rich regions are often generated by integration of TEs followed by RIPing [61,63,64]. Inactivated remnants of different TE families were found in most AT-rich regions (Appendix A) [28]. In *V. dahliae*, lineage-specific regions enriched in effector genes were formed by TE-triggered rearrangements, contributing to the evolution of pathogen virulence [65]. The genome of *Leptosphaeria maculans* is divided into alternating GC-equilibrated regions and AT-rich blocks containing TE inactivated by RIPing and enriched in effector genes [41]. In the genomes of *B. fabae* and other Sclerotiniaceae, no evidence for such genes next to AT-rich regions was found [30,61]. Nevertheless, the role of TE and RIPing activity in enhancing genome plasticity in BfabG12 is supported by the observation of frequent losses of gene clusters next to AT-rich regions. Because RIPing is a premeiotic event, its occurrence indicates the existence of a sexual cycle in *B. fabae*, which has not yet been observed in nature. In addition to the large number of RIPed TE, several intact, presumably active TE copies have remained, and all but two of them are located next to AT-rich regions (Appendix A). In *B. cinerea*, recent integrations of Gypsy LTR-RE sequences into the promoter of a gene involved in efflux-mediated fungicide resistance have occurred [66].

Compared to BcB0510, the overall gene count is lower in BfabG12 (11420 vs. 11707). This difference is largely due to a higher number of DNA segments that were lost in BfabG12. Of the 426 BcB0510 genes that are missing in BfabG12, 308 form gene clusters. Since a substantial fraction (31.9%) of these losses occurred at chromosomal ends, recombination events involving chromosomal termini may have played an important role in the evolution of *B. fabae*. More than half (170) of the BcB0510 genes that are lost in BfabG12 are involved in SM biosynthesis. Increased chromosomal dynamics of SM clusters is well documented for *Botrytis* species [30], but the extent of losses of SM genes in *B. fabae* is remarkable for such closely related species. The gene clusters for the phytotoxins botrydial and botcinic acid are both located in AT-rich regions with relics of TE activity [40,52]. The botcinic acid cluster is located next to a telomere of Chr1 in BcB0510,and on a telomere of Chr10 in BfabG12. It occurs in intact or truncated versions in all but 1 of 18 sequenced Botrytis species [31]. Interestingly, the botcinic acid cluster has been lost in a genetically distinct subgroup of *B. cinerea* [9]. Parts of a cluster of 11 genes, including Bcpks11 and Bcnrps9 (Appendix A), most of which are missing in *B. fabae*, were probably obtained by horizontal gene transfer from *F. graminearum* [67]. Overall, this indicates a significantly reduced SM synthesis capability of *B. fabae*. Nevertheless, *B. fabae* seems to be able to produce two unique sclerotium-specific SM, and one SM that is induced during infection. In addition to gene losses, another significant factor in the reduced coding capacity of BfabG12 is a higher frequency of gene degeneration compared to BcB0510. To our knowledge, this phenomenon of preferential gene degeneration in one species has not been described before in other fungi. In 170 out of 401 cases, truncations of BfabG12 genes could be traced to single-point mutations leading to premature stop codons or loss of start codons. This indicates that these mutations have occurred recently during evolution.

Differentially occurring genes in the two species were analyzed for their transcriptional activity. Average expression levels of the strain-specific genes, including those truncated in the other strain, were markedly lower compared to the common genes, indicating that most of them might not have a major function. Sclerotium-specific expression was observed most frequently in both species, followed by in planta-specific expression. Sclerotia serve as resting structures and as recipients of spermatia during sexual development. They differ markedly in size between *B. fabae* and *B. cinerea*, which might explain the large number of species-specific genes showing sclerotium-specific development. An interesting observation was the identification of a *B. fabae*-specific, in planta-induced gene cluster that includes a fatty acid desaturase, an acetyl transferase, two cytochrome P450s and an efflux transporter, possibly involved in the synthesis of a fatty acid derivate (Figure 5B). Lipids containing desaturated fatty acids have been found in to play a role in signaling between mycorrhizal fungi and their host plants [68].

In all major functional categories, the number of BcB0510-specific genes largely exceeded the number of BfabG12-specific genes, such as those for oxidizing and reducing enzymes, hydrolases, transferases and transporters. Furthermore, 36 genes encoding transcription factors exist only in BcB0510, compared to only three BfabG12-specific genes. These differences also hold true for the comparison of species, and collectively indicate that *B. fabae* has a reduced metabolic and regulatory capacity. In contrast to most fungal pathogens, effector-like proteins that could explain host specificity have not yet been identified in *Botrytis* [30]. Several secreted proteins of *Botrytis* encode cell death-inducing proteins, but there is no evidence that they play a major role in host range determination [69,70,71]. Comparison of the in silico secretomes revealed few candidates for species-specific effector proteins with functional relevance. Two adjacent genes (Bfab11g05317/18; Appendix A) were found to be upregulated in BfabG12 during plant infection and are candidates for functional analysis. Of the *B. cinerea*-specific genes for secretory proteins, nine show significant upregulation during plant infection. In contrast to several plant pathogenic fungi, we did not observe regions enriched in effector genes in *B. cinerea* or *B. fabae*. In the related broad host range necrotroph *S. sclerotiorum*, a significant association between the location of secreted proteins and regions with a high RIPing index were observed, but no evidence for specific compartments with effector-like genes [61].

Regarding evolutionary aspects, our results demonstrate substantial losses of coding capacity of *B. fabae* after its separation from *B. cinerea*. Because the closest relative of both species, *B. pseudocinerea*, has a broad host range similar to *B. cinerea*, it seems most likely that the three species have a broad-host-range progenitor. Therefore, the loss of functional genes could have played an adaptive role for *B. fabae* during its evolution towards a pathogen with a narrower host range of just a few genera of legumes. Conversely, several of the *B. cinerea* genes absent in *B. fabae* could be required for its wide host range. Interestingly, 150 of 185 genes present in all *B. cinerea* strains and missing in the three *B. fabae* strains were found in the draft genome of *B. pseudocinerea* (Appendix A). These genes are candidates for contributing to the extended host range of *Botrytis*. Our microscopic data indicate that necrotrophic growth of *B. fabae* differs from that of *B. cinerea* by fewer plant cells that are killed in front of the expanding hyphae (Figure 1). This could indicate that the secretome of *B. fabae* has a lower phytotoxic activity. To further evaluate the secretomes of the two species, inoculation droplets were collected 48 h after inoculation and analyzed by MS/MS. As shown previously, this is a simple but sensitive method for detailed analysis of the proteins that are secreted during the infection process [25,32]. Qualitative comparison of the proteins secreted during infection revealed a high degree of similarity, but also a higher number or proteins that were exclusively found or enriched in *B. cinerea*. Interestingly, 17 phytotoxic proteins that were detected in the *B. cinerea* secretome [23,69] were also found in *B. fabae*. These data indicate that the host-specific *B. fabae* uses a similar set of CDIPs as *B. cinerea* to attack its preferred host *Vicia faba*, though some CDIPs were found in lower amounts. Therefore, the secretomes of the two species do not seem to differ greatly in their phytotoxic potential. This might be unexpected, based on the assumption that the large diversity of phytotoxic compounds produced by *B. cinerea* is part of the multi-host necrotrophic strategy towards a large diversity of host plants. Although *B. fabae* infects only a small number of host plants, it seems to follow a similar infection strategy.

This study has focused on differences in gene counts between *B. fabae* and *B. cinerea*. However, phenotypic diversity between species can also be caused by different functions and regulatory properties of homologous proteins. In a comparative study of a similar wide and narrow host range pair of closely related fungal pathogens, *S. sclerotiorum* and *S. trifoliorum*, transcriptomic data revealed that *S. sclerotiorum* has a higher degree of transcriptional plasticity during infection of different host plants, indicating a higher regulatory adaptability [72]. The availability of complete genome sequences has laid the foundation for the elucidation of the molecular basis for the phenotypic diversity of *B. fabae* and *B. cinerea*. To achieve this goal, functional studies involving the generation and characterization of mutants in the differentially occurring genes, as well as their overexpression or transfer into the other species, are required to identify those genes that contribute to infection on one or multiple host plants.

## Figures and Tables

**Figure 1 jof-10-00216-f001:**
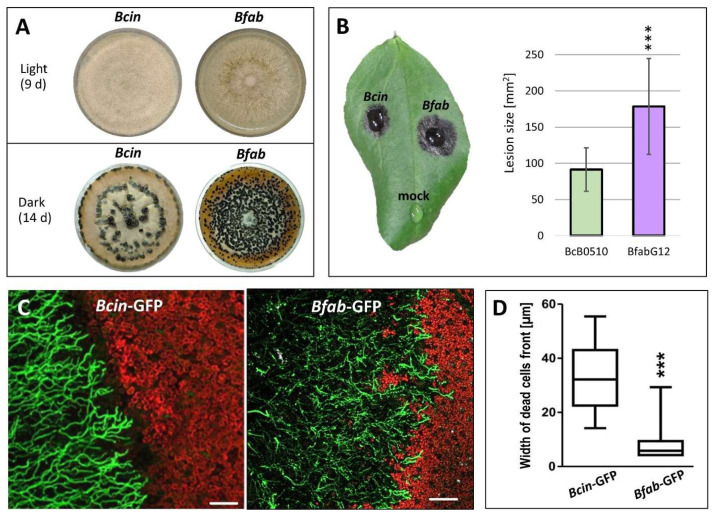
Comparison of growth and infection of *B. cinerea* and *B. fabae*. (**A**) Colonies (4 d.p.i.) formed in the light and sclerotia (14 d.p.i.) in the dark. (**B**) Lesions formed on *Vicia faba* leaf. (**C**) Fluorescence micrograph of expanding lesions (3 d.p.i.) formed by GFP-labeled *B. cinerea* and *B. fabae*. Scale bar: 50 µm. Note loss of red chlorophyll autofluorescence around spreading hyphae of *B. cinerea*, but to a lesser extent around *B. fabae* hyphae. (**D**) Quantification of host cell death ahead of expanding *B. cinerea* and *B. fabae* hyphae. ***: Significance: *p* < 0.001.

**Figure 2 jof-10-00216-f002:**
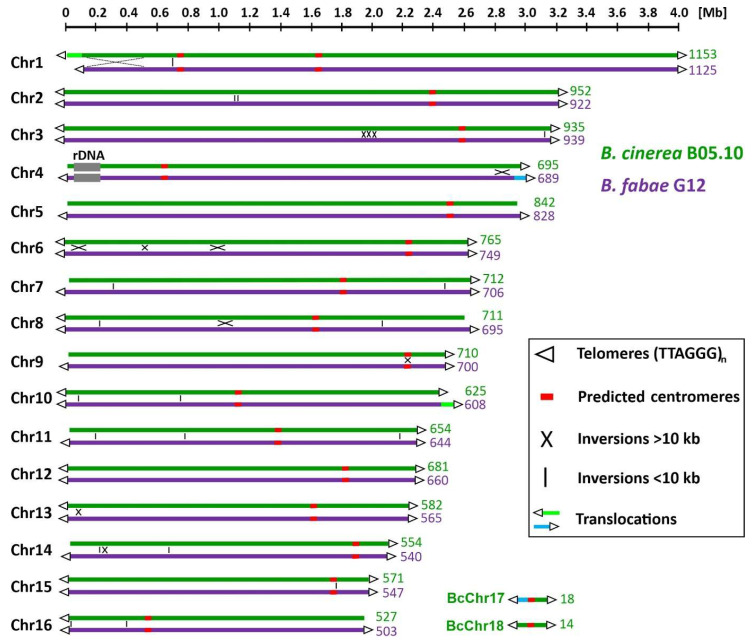
Comparative chromosomal maps of *B. cinerea* strain B05.10 and *B. fabae* strain G12. The number of genes on each chromosome is indicated.

**Figure 3 jof-10-00216-f003:**
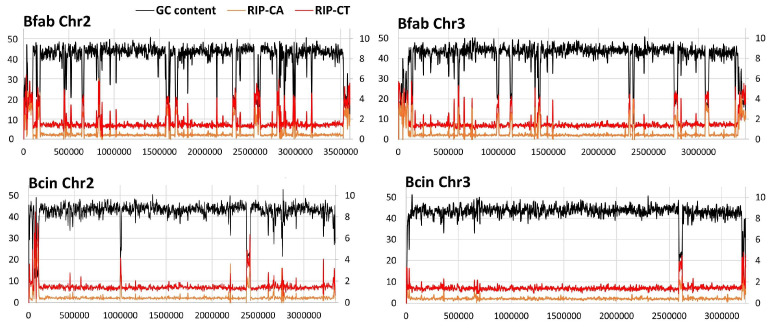
Comparative maps of *B. fabae* G12 and *B. cinerea* B05.10 chromosomes 2 and 3, highlighting different GC contents and RIP index values.

**Figure 4 jof-10-00216-f004:**
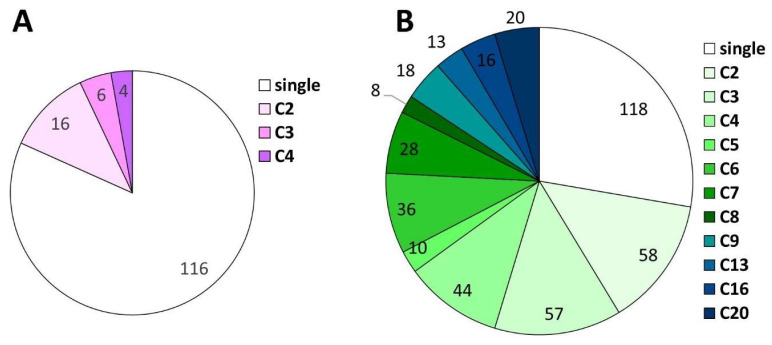
Extent of clustering of strain-specific genes in *B. fabae* G12 (**A**) and *B. cinerea* B05.10 (**B**). Numbers of genes in the clusters (C2, C3 …) are shown and highlighted in different colors.

**Figure 5 jof-10-00216-f005:**
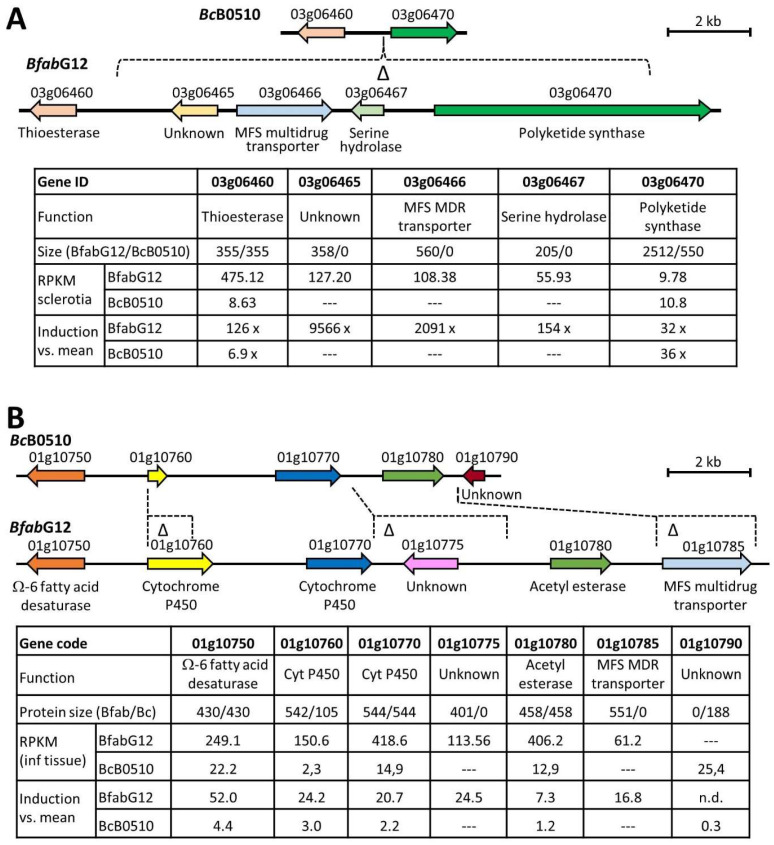
Two *B. fabae*-specific gene clusters showing stage-specific expression. (**A**) Cluster including a polyketide synthase gene with sclerotia-specific expression. (**B**) Cluster possibly involved in the synthesis of a fatty acid derivative showing in planta-induced expression. Below the maps, predicted sizes of encoded proteins, transcript expression values (RPKM), and expression levels relative to mean expression in the other growth stages (“induction vs. mean”) are shown.

**Figure 6 jof-10-00216-f006:**
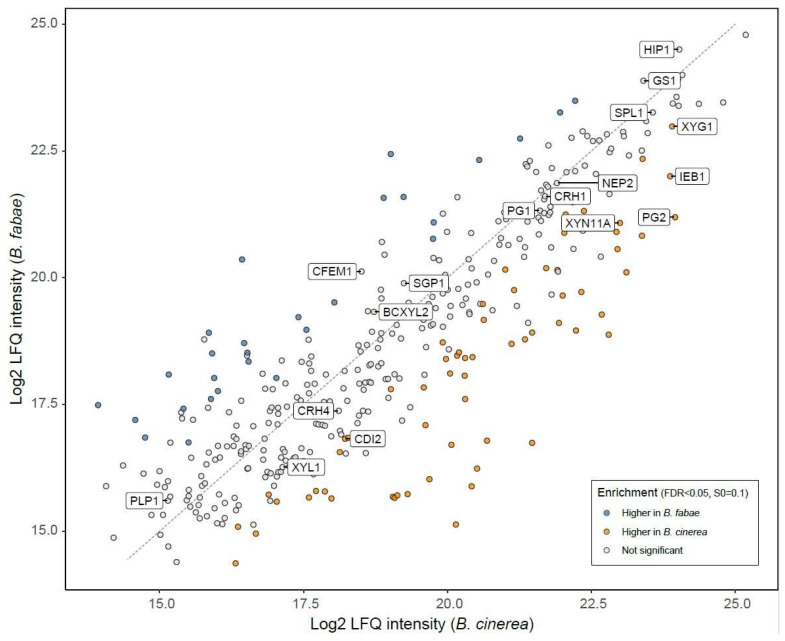
Scatterplot comparing intensities of secreted proteins of *B. cinerea* B05.10 and *B. fabae* G12 on *Vicia faba* leaves (48 h.p.i.). CDIPs are shown in boxes (see Appendix A for references).

**Table 1 jof-10-00216-t001:** Chromosome and genome sizes and GC contents of *B. fabae* G12 and *B. cinerea* B05.10.

Chrom.	*B. fabae* G12				*B. cinerea* B05.10			
	Size (Mb)	% GC	AT-Rich Regions ≥ 1 kb		Size [Mb]	SM% GC	AT-Rich Regions ≥ 1 kb	
			Number	Size (kb)			Number	Size (kb)
1	4.17	41.42	27	417.7	4.11	42.63	21	179.9
2	3.60	40.03	45	522.8	3.34	42.44	16	135.2
3	3.47	41.02	39	368.5	3.23	42.96	8	76.4
4	2.55 + 0.58 *	41.10	18	246.9	2.47 + 0.58 *	42.29	5	88.7
5	3.03	41.63	21	247.0	2.96	42.95	8	70.9
6	2.97	39.27	32	499.4	2.73	41.80	16	144.9
7	2.84	40.64	31	311.3	2.65	41.97	10	122.3
8	3.00	38.26	35	626.6	2.62	42.08	23	134.9
9	2.91	38.63	50	543.5	2.55	41.95	15	117.0
10	2.70	39.30	27	457.3	2.42	42.28	8	102.4
11	2.37	41.09	19	219.0	2.36	41.65	20	141.1
12	2.44	40.23	29	307.8	2.35	41.94	14	112.4
13	2.29	40.05	22	299.1	2.26	41.51	23	133.8
14	2.20	39.25	30	379.8	2.14	41.96	27	125.5
15	2.21	39.07	28	402.7	2.03	41.59	18	152.7
16	2.17	38.09	30	455.4	1.97	41.30	13	126.8
17	-	-	-	-	0.25	27.28	20	146.9
18	-	-	-	-	0.21	28.05	16	119.6
Total	45.50	40.02	483	6304.8	43.23	42.00	281	1964.9

* Estimated sizes of rDNA repeats [31].

**Table 2 jof-10-00216-t002:** Gene counts of *B. fabae* G12 and *B. cinerea* B05.10.

	BfabG12	BcinB05.10
A: Total no of genes	11,420	11,707
B: Strain-specific genes *	129 (148 *)	426 (471 *)
C: Genes truncated in other strain *	61 (72 *)	359 (401 *)
D (B + C): Strain-specific intact genes	190	785
expressed (RPKM > 2)	116	532
with annotation **	54	407
genes in clusters of ≥2 genes	32	335
unique genes ***	17	83

* Including truncated genes; ** proteins with predicted functions or conserved domains; *** genes missing in other *Botrytis* spp.

**Table 3 jof-10-00216-t003:** Presence and absence of confirmed and predicted secondary metabolite gene clusters in *B. cinerea* B05.10 and *B. fabae* G12 (cf. Appendix A). PKS: polyketide synthase; NRPS: nonribosomal peptide synthases; STC: sesquiterpene cyclase; DTC: diterpene cyclase; DTS: dimethylallyl tryptophan synthase. Red background: Secondary metabolite synthesis probably abolished. Pink background: Secondary metabolite synthesis questionable.

No	Type of Metabolite	Genes	Key Enzyme(s)	Presence or Absence of Genes		Stage-Specific Expression	
				Bcin B05.10	Bfab G12	Bcin B05.10	Bfab G12
1	Boticinic acid	13	PKS + PKS	+	+	Inf. leaves	Inf. leaves
2	Abscisic acid	5	Sesquiterpene cyclase	+	3 del	Inf. leaves	
3	Siderophore	1	NRPS	+	+		
4	Retinal (carotenoid)	2	Squalene synthase	+	+		
5	Diterpene	2	Diterpene cyclase	+	+	Inf. leaves	Inf. leaves
6	Peptide	5	NRPS	+	+		
7	Polyketide	5	PKS	+	+		
8	Polyketide	5	PKS	+	PKS trunc	Sclerotia	Sclerotia
9	Peptide	2	NRPS	+	NRPS trunc	Sclerotia	
10	DHN melanin	7+2	PKS + PKS	+	+		
11	Polyketide	11	PKS	+	1 trunc		Sclerotia
12	Polyketide	20	PKS	+	1 trunc		
13	Polyketide	10	PKS	+	PKS trunc, 5 del	Sclerotia	
14	Polyketide	5	PKS	PKS trunc, 3 del	+	(Sclerotia)	Sclerotia
15	Diterpene	1	Diterpene cyclase	DTC del	+		
16	Polyketide	1	PKS	+	1 del		
17	Peptide	6	NRPS	+	+	Sclerotia	Sclerotia
18	Sesquiterpene	1	Sesquiterpene cyclase	+	+		
19	Indole-diterpene	1	Polyprenyl synthetase	+	2 trunc + 1 del		
20	Polyketide	7	PKS	+	1 trunc, 4 del	Sclerotia	(Sclerotia)
21	Polyketide	1	PKS	+	PKS del	Inf. leaves	
22	Polyketide	1	PKS	+	+		
23	Polyketide	6	PKS	+	PKS + 5 del	Sclerotia	
24	Sesquiterpene	2	Trichodiene synthase	+	+		
25	Diterpene	5	Diterpene cyclase	1 trunc	DTC + 2 trunc		
26	Polyketide	4	PKS	PKS + 2 del	+		
27	Polyketide	1	PKS	+	PKS del		
28	Peptide-polyketide	9	NRPS + PKS	+	+	Inf. leaves	Mycelium
29	Siderophore	2	NRPS	+	+		
30	Peptide	3	NRPS	+	+	Sclerotia	
31	Botrydial	7	Sesquiterpene cyclase	+	STC + 5 del	Mycelium	
32	Polyketide	5	PKS	+	2 trunc		
33	Oxalic acid	1	Oxaloacetate hydrolase	+	+	Inf. leaves	Inf. leaves
34	Pyrones etc.	1	Chalcone synthase	+	+		
35	Sesquiterpene	5	Sesquiterpene cyclase	1 trunc	1 del		Inf. leaves
36	Polyketide	5	PKS	+	PKS trunc, 3 del	Sclerotia	(Sclerotia)
37	Peptide-polyketide	11	NRPS + PKS	+	NRPS + PKS + 5 del, 2 trunc		
38	Alkaloid	2	Dimethylallyl trp synthase	+	+		
39	Polyketide	6	PKS	+	4 del, 1 trunc		
40	Alkaloid	2	Dimethylallyl trp synthase	+	DTS trunc		
41	Siderophore	2	NRPS	+	+	Inf. leaves	
42	Polyketide	3	PKS	+	1 trunc		
	Total no of genes	170		7 del, 3 trunc	43 del, 18 trunc		

**Table 4 jof-10-00216-t004:** Classification of functionally annotated, strain-specific genes in *B. fabae* G12 and *B. cinerea* B05.10.

Functional Classification	BfabG12- Specific	BcB0510- Specific
SM metabolite key enzymes	2 (1 */1 **)	13 (7 */6 **)
Cytochrome P450 monooxygenases	7 (3/4)	22 (14/8)
Oxidoreductases, oxygenases	4 (3/1)	72 (43/29)
Ankyrin, WD40 or TPR repeats proteins	2 (1/1)	26 (7/19
HET proteins	4 (1/3)	19 (6/13)
Transporters	4 (3/1)	34 (14/20)
Transcription factors	3 (1/2)	36 (17/19
Signal transduction proteins	1 (0/1)	4 (1/3)
Hydrolases	2 (1/1)	52 (30/22)
Transferases	3 (1/2)	31 (13/18)
CAZymes	2 (1/1)	27 (15/12)
Secreted proteins ***	12 (6/6)	31 (20/11)
DNA/RNA-modifying enzymes	4 (2/2)	11 (7/4)
Proteins with other functions	4 (2/2)	29 (12/17)
Sum	54 (26/28)	407 (206/201)

* Missing in the other strain; ** Truncated in the other strain; *** Excluding CAZymes, including hypothetical proteins.

## Data Availability

The genome sequencing data of *B. fabae* G12 are available at NCBI Datasets as GenBank assembly GCA_032594075.1. All other data, including fasta files of all BfabG12 mRNAs, CDS and proteins, are contained within the article and Appendix A.

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
