# Peer review of "Genome Comparisons between Botrytis fabae and the Closely Related Gray Mold Fungus Botrytis cinerea Reveal Possible Explanations for Their Contrasting Host Ranges"

_jof, 2024, doi:10.3390/jof10030216_

Round 1

Reviewer 1 Report

Dear Authors,

I have had the privilege of reviewing your manuscript entitled "Comprehensive Genome Analysis of Botrytis fabae: Insights into Host Specificity and Evolution," which delves into the genetic underpinnings that distinguish Botrytis fabae's host specificity from the broad host range of Botrytis cinerea. Your work is a significant contribution to the field of phytopathology, offering deep insights into the evolutionary mechanisms that drive host specificity in fungal pathogens.

Your introduction effectively sets the stage by highlighting the economic and ecological relevance of understanding fungal pathogen evolution. However, a more detailed discussion on the impacts of both B. fabae and B. cinerea, coupled with a review of existing literature on fungal host specificity, could provide a richer context for your study. In the materials and methods section, the use of cutting-edge sequencing technologies and thorough genomic analyses stands out as a robust approach. Nonetheless, incorporating additional comparative or protein-interaction analyses could further elucidate the functional implications of the genomic differences observed.

The results section provides compelling evidence of the genetic mechanisms underlying host specificity in B. fabae, notably through gene loss and degeneration. While these findings are significant, a more detailed functional analysis of identified genes and proteins could enhance understanding of their roles in pathogenicity. The discussion integrates your findings within the broader context of pathogen evolution and host interaction, offering valuable insights into the genetic basis of host specificity. Expanding this discussion to include potential agronomic implications and future research directions could increase the relevance and applicability of your work.

Before proceeding to the final decision, it is essential to note that the manuscript does not currently adhere to the formatting template required by the journal. Ensuring that your submission conforms to the journal's template in your next version will be necessary for publication consideration.

Overall, your manuscript is a substantial contribution to our understanding of fungal pathogen evolution and host specificity. It opens new avenues for research into the genetic factors that underpin pathogen-host interactions and offers potential strategies for crop protection and disease management.

As for revisions, I suggest minor revisions to strengthen your manuscript. First, expanding the introduction to include a broader review of the economic impacts and previous research on fungal host specificity would provide a more comprehensive background. Second, incorporating additional functional analyses or comparative studies could substantiate the link between genomic variations and pathogenicity. Lastly, a more detailed exploration of the agronomic implications of your findings in the discussion would enhance the practical relevance of your study.

Given the depth of your research, the minor nature of the suggested revisions, and the need to adjust the manuscript format to the journal's template, I believe your manuscript should be considered for publication following minor revision. Your work represents a meaningful advancement in the field of phytopathology and microbial evolution, and I anticipate that it will spark further research into the mechanisms of host specificity in fungal pathogens.

Sincerely,

No minor details are commented.

Author Response

Point 1:

First, expanding the introduction to include a broader review of the economic impacts and previous research on fungal host specificity would provide a more comprehensive background.

A: We have followed this suggestion and added a paragraph at the beginning of the introduction which provides some background about the current knowledge and economic impacts of fungal host specificity:

‘Plant diseases caused by fungi are among the greatest threats for for global crop production and human food supply. Plant pathogenic fungi are generally highly adapted organisms and often show high degrees of host specificity. For example, rust and powdery mildew fungi which belong to the economically most important plant pathogens, are completely dependent on their host for growth and reproduction, and often infect only one of few plant species. The molecular basis of host specificity is not yet well understood. An important role is played by secreted fungal effector proteins that promote infection by the pathogen. On the other hand, plants containing resistance proteins which recognize specific effectors can build up a strong immune response and prevent infection. In addition, the ability of some fungi to release phytotoxic metabolites and to overcome the chemical defence of their host plant, are further determinants of infection which can also affect host specificity [1]. Understanding of the determinants of host specificity is of great agronomic im-portance, because evolutionary pressure can lead to sudden host jumps, as exemplified by the appearance in the 1980s in Brazil of a wheat pathotype of the rice blast fungus Magnaporthe oryzae, which is causing increasing damage in many wheat producing countries [2].’

Point 2:

Second, incorporating additional functional analyses or comparative studies could substantiate the link between genomic variations and pathogenicity.

A: The intention of this request is not clear to us. We have presented a very detailed and comprehensive comparative analysis of the two genomes regarding their structure and the predicted functions of the strain- and species-specific genes (see Figs. and Tables 3, 4, S4, S5, S7, S8, S11, S12). We have also discussed some predictions based on the differential occurrence and expression of genes in BfabG12 and BcB0510, such as their differential abilities to synthesize several secondary metabolites, and the larger regulatory capacity of B. cinerea based on the higher number of genes encoding transcription factors. Any functional studies to substantiate the link between genomic variations and pathogenicity are beyond the scope of this manuscript.  

To comply with the intention of the reviewer, we have added two sentences at the end of the discussion, which emphasize the need for functional studies:

‘The availability of complete genome sequences will greatly facilitate has laid the founda-tion for the elucidation of the molecular basis for the phenotypic diversity of B. fabae and B. cinerea. To achieve this goal, functional studies involving the generation and chracteriza-tion of mutants in the differentially occuring genes, as well as their overexpression or their transfer into the other species, are required to identify those genes that contribute to infec-tion on one or multiple host plants.

Point 3:

Lastly, a more detailed exploration of the agronomic implications of your findings in the discussion would enhance the practical relevance of your study.

A: We agree that it is always important, in particular for the public, to point out the potential practical relevance our research. The results presented in our manuscript, however, make it difficult to draw any solid conclusions regarding the practical implications of our study.

Reviewer 2 Report

This study compares the genomes of two closely related Botrytis pathogens. Although facilitated by the close relation between the two species, the approach and results are nevertheless a really beautiful application of long read genomic sequencing. The differences between the two species lead to testable hypotheses about the basis for the contrasting (wide vs narrow) host ranges. Although the manuscript is concise, it accesses a very large data set, including not only the genome sequence but also transcriptome and secretome. 

inoculation droplets - this looks like new way to obtain secreted proteins, rather than delicate and low-yield centrifugations used to enrich soluble proteins in the host plant apoplast; though the authors' 2018 New Phytologist paper is cited, it might be appropriate to emphasize the novelty and simplicity (unless it is already standard and I'm just not familiar with it?) of the technique (and what might be missed or selectively enriched).

lines 379 and 539 BfG12 should be BfabG12

Author Response

Point 1:

Does the title describe the article's topic with sufficient precision?

No: Genome sequence comparisons can't reveal evidence for contrasting host ranges, because the host ranges are already known. So, what is revealed? A possible genomic basis or explanation for the contrasting host ranges. Perhaps the authors can consider something like the following: Sequence comparisons between Botrytis fabae and the closely related grey mould fungus Botrytis cinerea reveal a genomic basis for their contrasting host ranges

A: We agree with this thoughtful comment about the title of our manuscript. We have changed the title, which now reads:

‚Genome comparisons between Botrytis fabae and the closely related grey mould fungus Botrytis cinerea reveal possible explanations for their contrasting host ranges‘

Point 2:

inoculation droplets - this looks like new way to obtain secreted proteins, rather than delicate and low-yield centrifugations used to enrich soluble proteins in the host plant apoplast; though the authors' 2018 New Phytologist paper is cited, it might be appropriate to emphasize the novelty and simplicity (unless it is already standard and I'm just not familiar with it?) of the technique (and what might be missed or selectively enriched).

A: Indeed we have already introduced the use of inoculation droplets for secretome analyses of infected tissue, both in Müller et al. (2018) and later in Leisen et al. (2022). However, we appreciate this comment and have modified the text to highlight the simplicity and sensitivity of this method (discussion, l. 568-571) Inspired by the suggestion of reviewer 2, we are planning to write a methods article about the potential of on planta secretome generation and analysis for the investigation of pathogenic plant microbe interactions.

lines 379 and 539 BfG12 should be BfabG12: Corrected in the text.